# Harnessing T Cells to Target Pediatric Acute Myeloid Leukemia: CARs, BiTEs, and Beyond

**DOI:** 10.3390/children7020014

**Published:** 2020-02-17

**Authors:** Rebecca Epperly, Stephen Gottschalk, Mireya Paulina Velasquez

**Affiliations:** 1Department of Oncology, St. Jude Children’s Research Hospital, 262 Danny Thomas Place, Memphis, TN 77030, USA; rebecca.epperly@stjude.org; 2Department of Bone Marrow Transplantation and Cellular Therapy, St. Jude Children’s Research Hospital, 262 Danny Thomas Place, Memphis, TN 77030, USA; stephen.gottschalk@stjude.org

**Keywords:** acute myeloid leukemia, immunotherapy, chimeric antigen receptor, CAR, bispecific antibodies, BiTE, DART

## Abstract

Outcomes for pediatric patients with acute myeloid leukemia (AML) remain poor, highlighting the need for improved targeted therapies. Building on the success of CD19-directed immune therapy for acute lymphocytic leukemia (ALL), efforts are ongoing to develop similar strategies for AML. Identifying target antigens for AML is challenging because of the high expression overlap in hematopoietic cells and normal tissues. Despite this, CD123 and CD33 antigen targeted therapies, among others, have emerged as promising candidates. In this review we focus on AML-specific T cell engaging bispecific antibodies and chimeric antigen receptor (CAR) T cells. We review antigens being explored for T cell-based immunotherapy in AML, describe the landscape of clinical trials upcoming for bispecific antibodies and CAR T cells, and highlight strategies to overcome additional challenges facing translation of T cell-based immunotherapy for AML.

## 1. Introduction

Despite advances in therapy, prognosis continues to be poor for patients with acute myeloid leukemia (AML) [1]. Targeted immunotherapy has the potential to improve outcome for this patient population while avoiding the long term toxicities associated with conventional chemotherapy. CD19-directed therapies for pediatric acute lymphocytic leukemia (ALL) have generated impressive responses and led to United States Food and Drug Administration (FDA) approval [2,3,4]. However, advancing immunotherapeutic strategies for AML has been hindered by additional challenges such as overlapping antigen expression on AML blasts and healthy tissues, T-cell persistence, and an immunosuppressive microenvironment. 

There are several immunotherapeutic strategies that have been developed for AML such as monoclonal antibodies [5], checkpoint inhibitors [6], cancer vaccines, natural killer cell add-back [7], and T cell-based therapies [8,9]. In this review, we will focus on strategies that target T cells to AML blasts, specifically highlighting bispecific antibodies and chimeric antigen receptor (CAR) T cells (Figure 1). We will discuss identification of target antigens applicable across T cell-based immunotherapies, review current and upcoming clinical trials, and identify challenges for T cell-based immunotherapies in AML and strategies to address them. 

Bispecific antibodies are molecules with distinct recognition domains recognizing both a specific tumor antigen on the AML blasts and CD3 on resident T cells [10]. By activating T cells and bringing them in contact with blasts at the immunologic synapse, they induce anti-leukemic cytotoxicity. In contrast, CAR T cells are generated by collecting T cells from a patient, genetically engineering them to express a CAR recognizing a specific tumor antigen, expanding the T cells ex vivo, and infusing them back into the patient [11]. Chimeric antigen receptors consist of an antigen recognition domain, traditionally from the single chain variable fragment of an antibody, hinge and transmembrane components, costimulatory domains, and an activation domain derived from the CD3ζ portion of the TCR [11]. While initial clinical experience has been primarily in adult patients with AML, clinical trials for pediatric patients are becoming available (Table 1, Table 2).

## 2. Identifying Target Antigens

Ideal antigens for cell-based immunotherapy are those that are expressed at high levels on malignant cells and absent or at low levels on normal tissues. Because of the challenge of identifying these differentially expressed targets, integrated screening efforts have been used in order to determine candidate antigens for targeted immunotherapy in AML [12]. Because of the relative heterogeneity of AML and overlap with hematopoietic progenitor cells or mature myeloid cells, it is likely that combinatorial therapies or advanced design techniques will be necessary to advance targeted T cell-based immunotherapy for AML. Most bispecific antibodies and CAR T cells currently being explored recognize antigens expressed on the cell surface only, which limits the pool of potential targets [10,13,14]. CD123 and CD33 are two of the antigens being explored currently as targets for bispecific antibodies and CARs (Table 1, Table 2). We discuss these along with other antigens currently under preclinical and clinical investigation. Strategies being translated to the clinic will be further discussed in Section 3 and Section 4.

### 2.1. Antigens with Overlapping Expression in AML Blasts, Leukemic Stem Cells and Normal Hematopoietic Progenitor Cells

CD123 or IL3Ra is a glycoprotein composed of the alpha subunit of the interleukin-3 receptor. It is widely expressed in hematologic malignancies including AML [15], both on differentiated leukemic blasts and leukemic stem cells [16], which makes CD123 an attractive immunotherapy target. A notable consideration for CD123 targeted immunotherapy is the concomitant low expression of CD123 on normal hematopoietic progenitor cells (HPCs), and mature cells of the myeloid lineage [17,18]. CD123 is also expressed at low levels on endothelial cells [19], which should lead to close monitoring for on-target off-tumor toxicity such as capillary leak syndrome (CLS). More recently, overexpression of CD123 has been described in high risk pediatric AML cases [20].

FMS-like tyrosine kinase 3 (FLT3), also known as CD135, belongs to the receptor tyrosine kinase class III family and is uniformly expressed on AML blasts, both those with wild type FLT3 and with internal tandem duplication (FLT3-ITD) [21,22]. Like CD123, FLT3 is also expressed on HPCs and other early progenitors [23].

### 2.2. Antigens Primarily Overlapping with Mature Hematopoietic Cells

CD33 is a transmembrane sialic acid-binding immunoglobulin-type lectin (SIGLEC) receptor expressed on AML blasts [24]. CD33 also has expression on hematopoietic progenitor cells, in addition to mature myeloid cells, which must be taken into account when considering on-target off-tumor toxicity [24]. CD33 is also expressed on myeloid-derived suppressor cells in the AML microenvironment, cells that promote an immunosuppressive microenvironment and dampen the effect of T-cell mediated killing [25]. Therefore, targeting CD33 has the potential to enhance anti-leukemic activity not only by direct cytotoxicity, but also by generating a more favorable microenvironment to promote T-cell activity. In addition to hematopoietic cells, CD33 is expressed on hepatic Kupffer cells [24], raising concerns for its role in hepatic toxicity and sinusoidal obstruction syndrome with previously used CD33-directed therapies [26].

C-type lectin-like receptor 1 (CLL1, also CLEC12A) is highly expressed in AML. CLL-1 is absent from hematopoietic progenitor cells but is present on mature myeloid cells [27], and is a potential therapeutic target.

CD70 is an immune checkpoint molecule identified on AML blasts, in addition to traditional antigen presenting cells [12,28]. CD70 has recently been described as a CAR T-cell target for AML, showing promising preclinical activity [29,30].

While the antigens mentioned in Section 2.1 and Section 2.2 are attractive given their wide expression on AML blasts, clinical targeting of these antigens requires thoughtful combinatorial therapies to avoid excessive myelotoxicity. Potential strategies for clinical application include bridging to hematopoietic stem cell transplant or additional engineering strategies.

### 2.3. Antigens Present on Multiple Tumor Types

Folate receptors are upregulated on an array of malignancies, both on the surface of tumor cells and on surrounding stroma and tumor-associated macrophages [31]. The folate receptor beta (FRβ) isoform is expressed on AML blasts [31,32,33]. FRβ CAR T-cell therapy has been investigated preclinically for AML and pediatric solid tumors [32,33,34].

NKG2D is a naturally occurring receptor present on NK cells and some T-cell subsets, which recognizes antigens on the tumor surface. In order to capitalize on this tumor specificity, NKG2D receptors have been used as antigen recognition domains for CAR and other T-cell engaging modalities [35,36,37].

Lewis Y is a tumor-associated antigen expressed on hematologic malignancies including AML and multiple myeloma, in addition to several epithelial solid tumors [38,39], and has been investigated as a target for CAR T-cell therapy [40].

### 2.4. Epitope-Specific Antigens

CD44 is a ubiquitously expressed membrane glycoprotein, which is overexpressed on several hematologic and solid malignancies but also healthy tissues such as lung, kidney, and gastrointestinal tract [41]. However, the CD44v6 isoform noted in AML and multiple myeloma is relatively tumor restricted [41], making isoform-specific targeting a potential immunotherapeutic strategy.

CD43 is a sialomucin transmembrane molecule universally expressed on leukocytes [42]. CD43s is a unique sialylated epitope, overexpressed in AML and only weakly expressed on normal myeloid cells [42]. A bispecific T-cell engaging antibody utilizing CD43s to selectively target AML blasts while sparing normal leukocytes has shown preclinical activity [43].

### 2.5. Antigens Present in Distinct AML Subsets

CD7 is a transmembrane glycoprotein molecule present on thymocytes and mature T cells that is important for T-cell interactions and differentiation [44]. CD7 is highly expressed in T-ALL [45,46]. It is also expressed in approximately 30% of patients with AML and has been correlated with low expression of wild type CEBPA and a worse prognosis [47,48,49,50,51,52,53], making it an attractive immunotherapeutic target. However, because it is also expressed on most mature T cells, additional engineering and processing techniques are necessary to avoid fratricide when using T cell-based therapies to target CD7 [54].

Leukocyte immunoglobulin-like receptor-B4 (LILRB4) is a protein highly expressed in monocytic AML (formerly M5 according to the French-American-British classification), a subset accounting for 20% of pediatric cases [55]. Preclinical studies of CAR T cells targeting LILRB4 have shown antitumor efficacy without toxicity against hematopoietic progenitor cells [56].

CD19 is a B-cell marker associated with B-ALL, that is also expressed in a small subset of patients with AML and mixed-phenotype acute leukemia, associated with t(8;21), who may benefit from CD19-directed therapies [57].

### 2.6. Intracellular Antigens

PR1 is an HLA-A2 restricted AML nonapeptide derived from neutrophil elastase (NE) and proteinase-3 (P3), which offers AML specificity but is only present in patients expressing HLA-A2 [58]. A bispecific T-cell engaging antibody has been developed targeting the PR1/HLA-A2 complex, which has shown preclinical efficacy and may be clinically applicable for a subset of patients [59].

Wilms Tumor 1 (WT1) is a zinc-finger transcription factor, which is an intracellular tumor-associated antigen which has limited low-level expression on normal tissues but is overexpressed in many hematologic and solid malignancies [60]. WT1 is overexpressed in AML, particularly in patients with poor prognosis. Because of its intracellular localization, WT1 cannot be targeted with traditional CARs or bispecific antibodies. However, WT1-derived peptides presented by HLA molecules can be targeted with T-cell receptors (TCRs) or CARs and bispecific antibodies that recognize HLA/WT1 peptide complexes [60,61].

## 3. Bispecific Antibody Clinical Development

Bispecific antibodies are antibodies that in general terms redirect an immune cell to a cancer cell by simultaneously targeting one antigen expressed on cancer cells (e.g., CD123, CD19) and one on immune cells (e.g., CD3, CD16). They vary in size, kinetics, and activity depending on their structure [10]. Strategies incorporating the immunoglobulin fragment crystallizable (Fc) domain have a longer half-life, but risk attracting macrophages that can hinder interactions at the immunologic synapse [62]. Approaches that do not incorporate an Fc domain may allow for less hindrance at the synapse, but have a shorter half-life [10]. Two of the better known strategies for the latter group include bispecific engager molecules (BiTE) and dual affinity retargeting antibodies (DART). A BiTE is a small molecule containing two different antigen recognition sequences bound by a short, flexible linker [10,63]. A DART also incorporates two single chain variable fragments (scFv) joined by a disulfide bond, which impacts steric interactions at the immunologic synapse [64].

Optimization of bispecific antibodies is ongoing in order to improve effectivity of these products in clinical application. Current clinical trials utilizing these strategies are listed in Table 1.

Designs for CD3×CD33 under current preclinical and early clinical research include traditional BiTEs, full-length bispecific antibodies, and trivalent or tetramer products with additional antigen recognition domains [25,63,65,66,67,68,69]. Preliminary clinical results for AMV564, a CD3×CD33 tandem diabody showed antileukemic activity in three of the first nine evaluable patients with no dose limiting toxicities identified, and has been granted orphan drug status by the FDA (FDA) [70].

CD3×CD123 targeting strategies include full length antibodies and DARTs [64,71]. Flotetuzumab, a CD3×D123 DART, has shown clinical activity in adults, and is now the first bispecific antibody for AML being studied in pediatric patients (Table 1). In addition to CD33 and CD123, CD3×CLL1 bispecific antibodies have shown preclinical activity [62,72,73] and are being evaluated in an ongoing phase 1 study (NCT03038230). FLT3×CD3 bispecific antibodies have shown preclinical efficacy, in addition to FLT3 CAR T cells which is discussed in Section 4 [74].

## 4. CAR T Cell Clinical Development

CAR T cells harness the specificity of an antibody and the cytotoxic activity of a T cell to provided targeted antitumor activity. In addition to antigen selection as discussed in Section 2, variations in each of the other CAR components can also impact activity. Intrinsic T-cell factors and additional modifications to the CAR T cells can modulate the antitumor activity and impact persistence, as previously reviewed [75].

Several groups have shown preclinical efficacy of CD123-CAR T cells [76,77,78,79,80,81,82,83,84], and these constructs have transitioned to early clinical investigation with encouraging initial results (Table 2) [85]. This has resulted in orphan drug designation by the FDA for the Mustang Bio CD123-CAR MB-102, currently under investigation at City of Hope Medical Center. Given anticipated on-target off-tumor activity against HPCs, CD123-CAR T cell trials have incorporated additional measures to ensure safety, including incorporating bridge-to-transplant options and safety switches to “turn off” CAR T cells after desired activity to avoid toxicity to infused HPCs after bone marrow transplant [77,79].

CD33-CAR T-cell strategies have also shown efficacy in preclinical models and are being translated to clinical application [24,79,86,87,88,89,90]. Like CD123-CAR T cells, there is potential that these will require a bridge-to-transplant strategy in the case of toxicity against HPCs. In addition to safety switches [77], an alternative strategy being explored to circumvent the challenge of expression on normal myeloid cells is genetic inactivation of CD33 in hematopoietic cells is to combine CD33-targeting T cells with CD33- hematopoietic stem cell transplant, which has shown feasibility in non-human primate models and is under further clinical development [91].

CAR T-cells targeting NKG2D have shown preclinical efficacy [36,92]. NKG2D-CAR T cells were investigated in a phase 1 clinical trial including seven patients with AML and five patients with multiple myeloma and demonstrated safety but no disease response [35]. Lack of response was attributed to poor CAR T-cell expansion and persistence, and work is ongoing to modify NKG2D CAR T cells to improve persistence [35]. A case report of NKG2D-CAR T cells did show antitumor efficacy in a single patient [93].

A phase 1 study of CAR T-cells targeting Lewis Y demonstrated safety and in some patients transient anti-leukemic activity, but no durable response despite persistence of infused CAR T cells [40]. CLL1-CAR T cells have shown preclinical efficacy [27,94] in AML and is in ongoing clinical development. There are ongoing clinical studies in China targeting CLL1 in combination with other AML antigens [95].

A clinical trial of FLT3-CAR T cells is also underway based on promising preclinical data [74,96,97]. It has been shown preclinically that FLT3 surface expression can be increased by simultaneous administration of tyrosine kinase inhibitors. This in turn can potentiate the effect of FLT3-directed immune therapy when used in combination [97,98].

## 5. Additional T Cell-Based Immunotherapy Strategies

### 5.1. Bispecific T-Cell Engager Secreting Cells

Because administration of bispecific antibodies can be limited by their short half-life, systemic effects, and reliance on native donor effector cells, groups have explored strategies to engineer cells to secrete bispecific antibodies. For example, T cells secreting CD123- or CLL1-specific BiTEs (Engager T cells) have anti-AML activity in preclinical models [99,100]. One advantage of engineering T cells to secrete BiTEs is that these cells can be further genetically modified to express costimulatory molecules or chimeric cytokine receptors [99,101]. Additionally, investigators have explored the anti-AML activity of engineered mesenchymal stromal cells that secrete CD33xCD3 bispecific antibodies in preclinical models [102,103].

### 5.2. T Cell Receptor Engineered T Cells

To overcome the limitation of recognizing only surface antigens on tumor cells, an alternative strategy is to engineer a specific TCRs, which allows for the recognition of intracellular proteins on MHC [60]. Because this strategy uses an MHC-dependent recognition strategy, the products are HLA-specific and therefore each product will be applicable to only a subset of patients whose blasts express a given antigen. TCR therapy targeting WT1 has shown promise in preventing relapse after HCT for AML in adults [61].

### 5.3. Tumor-Associated Antigen Specific T Cells

Patients with AML have naturally occurring tumor-reactive T cells, though often without appropriate quantity or activity to control disease [104]. In contrast to direct genetic modification, it is possible to select for and expand donor-derived naturally occurring T cells against multi tumor specific antigens. This strategy is being explored in early clinical studies (NCT02494167) [105].

## 6. Challenges of Bispecific Antibody and CAR T-Cell Therapies

### 6.1. Toxicity

Bispecific antibodies and CAR T cells show promise in improving outcomes for patients with AML. However, there are additional challenges to overcome for successful clinical application. A comprehensive discussion of these challenges is beyond the scope of this discussion but have been recently reviewed as outlined. In addition to the on-target off-tumor toxicities discussed in Section 1, cytokine release syndrome and immune effector cell associated neurotoxicity syndrome have been described after CAR T infusion and appear to be generalized phenomena across multiple target antigens and tumor types [106]. Similar phenomena are described with bispecific antibodies [107].

### 6.2. Immune Escape

A major mechanism of relapse after targeted T cell-based therapies is antigen escape, which has been described with both CAR T cell and antibody-based CD19-directed therapy [108,109]. Given the heterogeneity of AML, this challenge is likely to be amplified with targeted therapy for AML [110]. One strategy to prevent antigen escape is targeting multiple antigens simultaneously to add selective pressure. Development of bispecific CAR T cells targeting CD123 and CD33 is ongoing [111].

### 6.3. AML Microenvironment

In AML, the immune suppressive microenvironment plays an important role in both primary therapeutic resistance and relapse in T cell-based therapies [112]. Immune suppressive cells including regulatory T cells [113,114] and myeloid-derived suppressor cells [115,116] among others can inhibit T-cell function and contribute to exhaustion. Anti-inflammatory cytokines including IL10 [117] and TGF-β [118] contribute to a suppressive environment, along with metabolic alterations [119,120]. Combination therapeutics and additional genetic modifications to T cells are potential strategies to modulate the environment to one that promotes anti-leukemic activity of modified T cells [59,121].

## 7. Conclusions

Targeted T cell-based immunotherapies offer great promise for AML therapy, but still face challenges in clinical application. Bispecific antibodies offer potential for use as an off-the-shelf product, though are limited by impact of half-life on administration and reliance on native host immune system for effector function. CAR T cells retain the specificity of antibodies with enhanced effector function, though there are additional challenges related to production and persistence of cellular therapy products. CD123 and CD33 stand out as promising antigens, though there are several others under investigation. Overcoming toxicities related to shared expression of target antigens on normal hematopoietic cells will be key in translating these therapies. Ongoing early clinical studies are being performed in patients with AML who have relapsed/refractory disease. However, as the field advances we will get a better understanding of the safety profile and efficacy of these therapies. This will allow devising clinical trials in the upfront setting, similar to CD19- and CD22-directed immunotherapies for ALL [122,123]. We are hopeful that synthesizing the lessons learned from the development of an array of antibody- and T cell-based targeted immunotherapies will help to swiftly advance the care for patients with AML going forward.

## Figures and Tables

**Figure 1 children-07-00014-f001:**
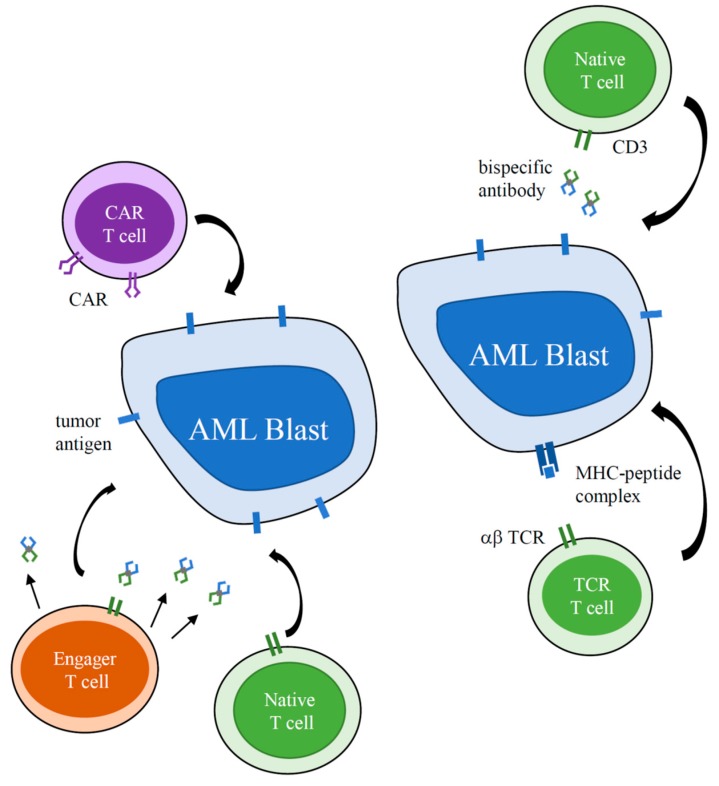
Strategies to Harness T Cells for Immunotherapy of acute myeloid leukemia (AML). CAR—chimeric antigen receptor, TCR—T-cell receptor, MHC—major histocompatibility complex.

**Table 1 children-07-00014-t001:** Bispecific Antibody Clinical Trials for AML.

Target	NCT	Institution/Sponsor	Product	Ages
CD123	NCT04158739	Children’s Oncology Group	*flotetuzumab (MGD006)*	<21
	NCT02715011	Janssen Research & Development, LLC	*JNJ-63709178*	18+
	NCT02152956	MacroGenics	*flotetuzumab (MGD006)*	18+
CD33	NCT02520427	Amgen	*AMG330*	18+
	NCT03144245	Amphivena	*AMV564*	18+
	NCT03915379	Janssen Research & Development, LLC	*JNJ-67571244*	18+
	NCT03516760	GEMoaB Monoclonals GmbH	*GEM333*	18+
CLEC12A (CLL1)	NCT03038230	Merus N. V.	*MCLA-117*	18+

Summary of active clinical trials according to www.clinicaltrials.gov as of 12/18/19.

**Table 2 children-07-00014-t002:** CAR T-cell clinical trials for AML.

Target	NCT	Institution/Sponsor	Ages
**United States**			
CD123	NCT02159495	City of Hope Medical Center	12+
	NCT03766126	University of Pennsylvania	18+
	NCT04109482	Mustang Bio	18+
	NCT03190278	Cellectis S. A.	18–64
	pending	St. Jude Children’s Research Hospital	<21
CD33	NCT03971799	Center for International Blood and Marrow Transplant Research (National Cancer Institute, Children’s Hospital of Philadelphia)	1–30
NKG2D	NCT04167696 NCT03018405 NCT02203825	Celyad	18+
FLT3	NCT03904069	Amgen	12+
**International**			
CD123	NCT03556982	Affiliated Hospital of the Chinese Academy of Military Medical Sciences, China	14–75
	NCT03796390	Hebei Senlang Biotechnology, China	2–65
	NCT04014881	Wuhan Union Hospital, China	18–70
	NCT03114670	Affiliated Hospital to Academy of Military Medical Sciences, China	18+
	NCT04106076	Cellectis S. A., United Kingdom	
CD7	NCT04033302	Shenzhen Geno-Immune Medical Institute, China	6 mos-75
CD44v6	NCT04097301	MolMed, Horizon 2020, Italy	I: 18–75 II: 1–75
Lewis Y	NCT01716364	Peter MacCallum Cancer Center, Australia	18+
CD19	NCT03896854	Shanghai Unicar-Therapy Bio-medicine Technology Co, Ltd., China	
CD123/CLL1	NCT03631576	Fujian Medical University, China	<70
CD123/CD33	NCT04156256	iCell Gene Therapeutics, China	child, adult
CCL1/CD33/CD123	NCT04010877	Shenzhen Geno-Immune Medical Institute, China	2–75
Muc1/CLL1/CD33/ CD38/CD56/CD123	NCT03222674	Shenzhen Geno-Immune Medical Institute, China	2–75
CD33/CD28/CD56/ CD123/CD117/CD133/CD34/MucI	NCT03473457	Zhujiang Hospital, China	6 mos+

Summary of active and completed clinical trials according to www.clinicaltrials.gov as of 12/18/19, in addition to upcoming trial at authors’ institution pending registration.

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
