# Peer review of "Harnessing T Cells to Target Pediatric Acute Myeloid Leukemia: CARs, BiTEs, and Beyond"

_children, 2020, doi:10.3390/children7020014_

Round 1

Reviewer 1 Report

The authors provide a comprehensive compilation of ongoing and future clinical trials investigating the efficacy of T cell based immunotherapy for AML. The introduction provided is sufficient and provides the required background and rationale for using different T cell based immunotherapy approaches such as CAR, BiTes, DARTs and naturally occurring tumor-specific T cells. The authors also outline the limitations of each therapeutic approach and ways currently being tested to overcome the same. The section outlining the currently targeted antigens and shedding light on their concomitant expression on normal cells along with ways investigated to overcome systemic toxicity (eg. CD33- bone marrow transplant, safety switches etc) is especially well written. In addition, the section on isoform-specific and modification-specific(sialylation) BiTE  is a valid inclusion to the review. 

Overall, the review is well written and makes a significant contribution to the field.  

Author Response

Thank you for the thoughtful comments.

Reviewer 2 Report

This is a comprehensive and well-written review on imunotherapy in AML. I would suggest to consider a small section on potential application in MRD positive patients, in whom immunotherapy is particularly acractive. 

Author Response

While initial  CAR T cell trials for AML are being undertaken in relapsed/refractory patients, we agree that T cell based immunotherapies for leukemia have shown efficacy in the MRD setting, particularly for bispecific antibodies in ALL. We have included a statement of where in therapy these strategies may be incorporated to our conclusion, including mention of efficacy in patients with low disease burden.